# Association of Preterm Birth and Exposure to Endocrine Disrupting Chemicals

**DOI:** 10.3390/ijms24031952

**Published:** 2023-01-19

**Authors:** Anish S. Kolan, Julianne M. Hall

**Affiliations:** Frank H. Netter MD School of Medicine, Quinnipiac University, NH-MED, Hamden, CT 06492, USA

**Keywords:** endocrine disrupting chemicals, preterm birth, reproductive system, bisphenol A, phthalates, pesticides, World Health Organization

## Abstract

Several studies in recent years have shown that endocrine disrupting chemicals (EDCs) can exert deleterious effects within several systems of the human body, such as the immune, neurological, and reproductive systems, among others. This review aims to summarize the investigations into the effect of EDC exposure on reproductive systems, namely preterm birth (PTB), and the efforts that international organizations have made to curb the harmful results of EDC exposure. To gather information, PubMed was initially searched for relevant articles containing the following terms: endocrine disrupting chemicals; preterm birth. PubMed was subsequently used to identify articles discussing the association between preterm birth and specific EDC exposures (BPA; phthalates; organochlorine pesticides; organophosphate pesticides; lead; PBDE; preterm birth). Both searches, limited to articles published within the past 20 years, identified several publications that have examined the association between various EDCs and PTB. While the findings of the studies differed, collectively they revealed sufficient evidence of a potential association between EDC exposure and risk of PTB. Thus, international organizations such as the United Nations Environmental Programme (UNEP) and World Health Organization (WHO) should continue to limit EDC exposure across the globe and monitor levels among individuals of reproductive age.

## 1. Background

EDCs are defined by the Endocrine Society as “exogenous chemicals, or mixtures of chemicals, that interfere with any aspect of hormone action [1].” They are often widely present in many communities and can alter various hormonal processes in the body. Some of the earliest reports of endocrine disruption occurred in the 1920s, when American pig farmers believed that swine herds fed moldy grain were less fertile than other herds [2]. The infertility was eventually attributed to mycoestrogens in the mold [2]. In the 1940s, Australian sheep farmers noticed infertility in sheep that grazed on certain fields of clover [2]. In this case, the infertility was due to phytoestrogens in the plants [2]. Despite these accounts, the idea of man-made chemicals disrupting animal and human physiology did not become popular until the 1962 publication of Rachel Carson’s book, *Silent Spring*, in which she discussed the potential consequences of pesticides and herbicides on wildlife populations [2]. Since then, evolving research has suggested that EDCs may be linked to reproductive, metabolic, neurologic, and carcinogenic effects [1].

Diethylstilbestrol (DES) is an EDC that has had a lasting legacy. It was prescribed to millions of women between 1940 and 1971 to prevent miscarriage [2]. Prescriptions were halted after evidence showed that daughters born to women who had taken DES during their pregnancies suffered vaginal cancers. The DES tragedy revealed the urgent need to address EDCs by showing the public that exposure can cause long-lasting damage in utero [2]. Thus, reproductive consequences of EDCs affect not only mothers, but also children, for the length of their lives. This review focuses on the literature examining whether in utero exposure to three categories [1] of EDCs (food contact materials, pesticides, chemicals in products) is associated with PTB, which the WHO defines as delivery prior to 37 completed weeks of gestation [3]. 

PTB is a problem of international concern, as it is the leading cause of death in children younger than 5 years old, and the frequency is increasing in countries with reliable data [3]. While the increasing rates may be due to better measurement, they are still concerning, considering the incredible improvements that have been made in medicine and technology. Every year, 15 million babies are born prematurely, and approximately 1 million children die annually due to PTB complications [3]. Those that survive are at greater risks of developing neurodevelopmental, growth, and respiratory disorders [4]. Of those 15 million premature deliveries, more than 60% occur in Africa and South Asia [3], underscoring the need for interventions in these regions. While PTB is a global issue, it affects low-income countries and families to a greater extent. In low-income countries, 12% of births are preterm, while in high-income countries, 9% of births are preterm. Within all countries, poor families are at greater risk of having a PTB [3]. Because PTB affects so many families globally, the WHO has provided wide-ranging guidelines to decrease rates. WHO recommends counselling on healthy diet and optimal nutrition, use of ultrasound to help determine gestational age and detect multiple pregnancies, a minimum of eight contacts with health professionals throughout pregnancy to identify and manage other risk factors and better access to contraceptives [3]. While these are all important and necessary goals, they can be difficult to implement, especially in low-income countries with larger underlying issues, such as lack of resources and access to adequate healthcare. As this review discusses, many studies have shown that mothers who delivered prematurely had elevated levels of EDCs in their urine, suggesting an association between EDCs and PTB (Table 1). If these studies continue to be corroborated in the future, it will be imperative that governing bodies enact policies restricting or prohibiting the use of EDCs to further combat PTB.

## 2. Parturition

Parturition, the process through which uterine contractions, cervical dilation, and cervical effacement leads to childbirth, is a complex process with many unknowns [18]. While fetal age can be estimated by multiple methods, clinicians measure the gestational age, which starts from the mother’s last menstrual period [18]. The average human gestational age is 40 weeks [18], but the American College of Obstetricians and Gynecologists (ACOG) defines a full-term pregnancy as one that ranges from 37 weeks to 41 weeks and 6 days [19]. As pregnancy progresses, several hormones facilitate the initiation of labor. The estrogen/progesterone ratio increases, and because estrogen stimulates uterine contractility while progesterone decreases contractility, this increasing ratio stimulates labor. Estrogen increases production of prostaglandins, such as prostaglandin F_2α_ and prostaglandin E_2_, which can also increase uterine contractility. Placental corticotropin-releasing hormone (CRH) and maternal serum CRH levels increase during late pregnancy and labor. Placental CRH stimulates secretion of fetal adrenocorticotropic hormone (ACTH). ACTH then increases fetal adrenal cortisol and fetoplacental estrogen production. This cortisol then stimulates placental CRH secretion, initiating a positive feedback loop. While CRH indirectly increases uterine contractions through the stimulation of cortisol production, it also directly promotes contractions by sensitizing the uterus to estrogen and progesterone. Oxytocin levels do not increase prior to labor, but its secretion from the posterior pituitary creates another positive feedback loop when it stimulates uterine contractions after cervical stretching [18].

## 3. Common PTB Causes

Just as many factors are involved in full-term parturition, various altered pathways can increase the likelihood of PTB. Studies have shown that an abnormal hypothalamus–pituitary–adrenal (HPA) axis, inflammatory responses, decidual hemorrhage, uterine and cervical abnormalities, and genetics can all increase chances of PTB [20].

The HPA axis can be activated by maternal and fetal stress, but uteroplacental ischemia has a higher association with PTB than maternal psychosocial stress does. The link between stressors and PTB can be attributed to elevated placental CRH, fetal ACTH, and glucocorticoid induction of the immunophilin co-chaperone FK506-binding protein-51 (FKBP51), which leads to progesterone withdrawal [20].

While inflammation is necessary to fight infections, an uncontrolled inflammatory response can have negative effects, including increasing susceptibility to PTB [20]. Studies have shown that genitourinary tract pathogens, chorioamnionitis, and absence of lactobacillus in the vaginal flora all increased the risk of PTB [20].

Vaginal bleeding can be a sign of decidual hemorrhage, a risk factor for PTB [20]. Decidual hemorrhage may be associated with PTB because decidual cells have a high concentration of tissue factor, a mediator in the coagulation process. Activation of tissue factors eventually leads to the formation of thrombin. In addition to serving a hemostatic role, thrombin increases myometrial contractions and inhibits progesterone receptor expression in decidual cells, both of which can facilitate PTB [20].

Physical changes to the uterus and cervix are well-described PTB risk factors [20]. Excessive uterine distension induces gap junction formation and upregulation of oxytocin receptors, inflammatory cytokines and prostaglandins, and myosin light chain kinase, all of which stimulate uterine contraction. Changes to the cervix, such as dilation and effacement unrelated to labor, may also increase chances of PTB [20]. 

Genetics may also play a role in PTB. Women who were born prematurely or have sisters who had PTB are at a higher risk of delivering their own children prematurely [20]. Additionally, multiple studies have highlighted single-nucleotide polymorphisms that are linked to increased risk of PTB [20].

Lastly, the ability of EDCs to harm maternal and fetal health by altering endocrine and reproductive function suggests that they may also increase the risk of PTB. Below, the association of exposure to three categories of EDCs (food contact materials, pesticides, chemicals in products) with PTB is objectively discussed in light of the available evidence.

## 4. Food Contact Materials 

Bisphenol A (BPA) and phthalates are used in plastic food storage containers to increase durability. Additionally, BPA is used in the epoxy-based linings of canned foods to prevent pathogen contamination, and phthalates are also found in personal care products. Because of increasing awareness about harm caused by excessive BPA exposure, several countries have banned its use in infant feeding bottles. However, BPA and phthalates continue to be used in many other products [1]. 

### 4.1. BPA

BPA is mostly used in the production of epoxy resins and polycarbonate plastics, which are used to make reusable food and drink containers, medical devices, water supply pipes, and many other products [5,6]. As these products age or are exposed to heat, BPA can enter the environment, through which humans mainly ingest it via skin contact, oral exposure, or inhalation [5,6]. While the potential for harmful consequences resulting from BPA exposure is widely agreed upon, its association with PTB has not been as clear, as both supporting and refuting evidence exist.

In their study of Mexican women, Cantonwine et al. compared the urinary BPA concentrations of women who delivered earlier than or at 37 weeks of gestation with those who delivered after 37 weeks [21]. Third-trimester spot urine samples from 30 participants who had full-term deliveries and 30 participants who delivered prematurely were randomly selected from a previous cohort study. In these samples, the authors found that urinary BPA concentrations of women who delivered prematurely were significantly higher compared to those of the women who delivered after 37 weeks. However, the authors noted limitations, such as a small sample size, single urine collection, and the use of participants’ last menstrual period to estimate gestational age [21].

In a subsequent study, Cantonwine et al. corrected these limitations by analyzing a larger sample of women longitudinally and using first-trimester ultrasounds to estimate gestational ages. This study consisted of 130 women who delivered before 37 weeks of gestation and 352 randomly selected women who delivered at or after 37 weeks of gestation. After analyzing three urine samples from all participants, the authors did not find a significant association between BPA levels and PTB [5]. However, they noted that after excluding indicated PTBs, the association between BPA levels and PTB strengthened. They chose to exclude these PTBs in their secondary analysis because these births are often induced and delivered early per obstetric guidelines, in order to reduce maternal and fetal complications. Additionally, their secondary analysis suggested that female fetuses may be at greater risk of PTB associated with BPA exposure compared to male fetuses. In terms of limitations, the authors acknowledged that their participant population may have restricted generalizability because they recruited from a referral center for high-risk pregnancies, so this population could have had a higher proportion of women with conditions that led to medically indicated protocol-driven PTBs [5]. 

Similar to the longitudinal Cantonwine et al. study, Huang et al. collected three urine samples from Chinese women, one during each trimester [22]. In this study, gestational ages were estimated using first-trimester ultrasounds. After analyzing the samples from 850 participants, the authors found that elevated BPA levels in all three trimesters were negatively associated with gestational age and positively associated with PTB. While the authors recruited many women, they reported only 21 PTB cases out of the 850 total participants. They believed the sample size, study design, and differences in biomarkers contributed to the low incidence of PTB reported [22].

While these studies show differing results, evidence warrants further research into the link between PTB and BPA exposure. Future studies should focus on recruiting a large sample pool, collecting multiple urine samples, and validating gestational age via ultrasound. Of the previously discussed studies, the longitudinal Cantonwine et al. study that found no significant association between PTB and BPA levels seemed to best meet these goals [5]. Though they did not find significant associations in their initial analysis, further investigation showed strengthened relationships and possible gender differences, highlighting the need for more research.

### 4.2. Phthalates

Similar to BPA, phthalates are found in many consumer products. Di-(2-ethylhexyl) phthalate (DEHP) exposure mainly occurs via consumption of food and water stored in containers containing DEHP. Other phthalates, such as benzylbutyl phthalate (BzBP), dibutyl phthalate (DBP), and diethyl phthalate (DEP), are often found in lotions, perfumes, and deodorants [7]. Currently, conflicting evidence regarding the effects of phthalate exposure and PTB exists, but with such widespread use, the impact of phthalates on PTB needs to be more clearly understood.

In a novel study, Zhang et al. examined a cohort of couples seeking fertility treatment in order to investigate the association between maternal and paternal preconception phthalate exposure and PTB [23]. The cohort yielded 420 births between 2005–2018. The use of in vitro fertilization protocol dates allowed the researchers to estimate gestational age with high accuracy. Similar to previously discussed studies, women provided one urine sample per trimester. However, they also provided samples once upon study entry and up to two samples per fertility treatment cycle. Male participants provided one sample at study entry and an additional sample per treatment cycle. The authors evaluated the levels of several phthalates but found increased risks of PTB with elevated levels of maternal preconception DEHP metabolites (∑DEHP) and cyclohexane-1,2-dicarboxylic acid monohydroxy isononyl ester (MHiNCH), a metabolite of 1,2-cyclohexane dicarboxylic acid diisononyl ester (DINCH). This is of note because use of DINCH has been increasing following the regulation of other phthalates. While maternal preconception phthalate levels were associated with increased risk of PTB, paternal preconception phthalate levels were not [23].

Ferguson et al. used a nested case-control design to analyze phthalate levels in three urine samples of 130 mothers who delivered prior to 37 weeks and 352 mothers who delivered at or after 37 weeks, the same cohort discussed in the Cantonwine et al. study [5,7]. They calculated levels of individual phthalates and the sum of DEHP metabolites (∑DEHP). Levels of mono-(2-ethyl)-hexyl phthalate (MEHP), mono-(2-ethyl-5-carboxypentyl) phthalate (MECPP), mono-n-butyl phthalate (MBP), and ∑DEHP were all higher in PTB cases compared to those in controls [7]. The authors believed that they strengthened their results by including a large sample size, multiple urine samples, and ultrasound-validated gestational ages, all of which previous studies not reporting a link between phthalate exposure and PTB lacked [7].

In a separate study, Ferguson et al. studied a cohort of Puerto Rican women [24], which is notable because Puerto Rico’s PTB rate of 11.5% is one of the highest rates in not only the US, but also the world. Additionally, the presence of many superfund waste sites exposes Puerto Ricans to several EDC pollutants. The data from this cohort supported this idea, as urinary concentrations of some phthalate metabolites were higher in this population compared to women of reproductive age from the National Health and Nutrition Examination Survey (NHANES) during the same period. Three urine samples from each of the 1090 participants were analyzed, and there were a total of 101 PTBs among the women. While average concentrations of DBP and di-isobutyl phthalate (DiBP) were associated with shorter gestational durations and increased odds of PTB, the study did not find a significant association between DEHP and PTB, in contrast to previous studies. The authors speculated that the lack of significance may be because Puerto Rican women are exposed to a low amount of DEHP, as its use has been decreasing in the US [24].

Recently, the association between phthalate exposure and PTB was examined using 16 cohorts of individuals living in the United States of America. Urinary phthalate metabolite analysis was used as a marker of phthalate exposure. The studies revealed that the odds of PTB was 12% to 16% higher in individuals with increased urinary concentrations of MBP, mono-isobutyl phthalate, mono(2-ethyl-5-carboxypentyl) phthalate, and mono(3-carboxypropyl) phthalate. Furthermore, the authors estimated that hypothetical reductions in phthalate levels by 10%, 30%, and 50% would have prevented 1.8, 5.9, and 11.1 of the approximately 90 preterm births per 1000 live births in the study population, respectively [25].

In summary, the published evidence discussed above revealed significant associations between phthalate exposure and PTB [7,23,24,25]. Future studies may focus their efforts on which phthalates increase risk of PTB, especially those that are being increasingly used as substitutes for other harmful chemicals, and how the timing of exposure affects PTB risk. In the study of Puerto Rican women, the greatest associations between DBP and DiBP exposure and PTB were among samples collected in the middle and late stages of pregnancy [24]. Together, these studies provide strong evidence that phthalate exposure, especially later in pregnancy, increases PTB risk, but future studies can provide more specific information about this relationship.

## 5. Pesticides

Two common classes of pesticides, organochlorines (OCPs) and organophosphates (OPPs), are used to control insects in agriculture and homes throughout the world [1]. These chemicals are established endocrine disruptors that manifest deleterious effects through altering neuroendocrine transmission in not only insects but also humans [8]. With a considerable proportion of the world’s population working in agriculture, understanding the link between pesticide exposure and PTB could provide an avenue to decrease PTB prevalence across the world.

### 5.1. Organochlorine Pesticides (OCPs)

OCPs are commonly used pesticides that are exposed to humans regularly because they have long half-lives and low biodegradability [10]. One of the most well-known OCPs, dichlorodiphenyltrichloroethane (DDT), gained popularity as a cheap pesticide that could be widely used in agriculture. After its harmful effects became known, the US banned DDT use in 1972 [2]. However, because of the long half-life of DDT, citizens of countries that enacted this ban continue to be exposed. To date, several countries have banned DDT. However, many others still use it, especially those with a high malaria prevalence [9]. Humans are most likely to be exposed to OCPs through foods, but exposure can also occur through breathing near or touching products that are contaminated [9]. Because OCPs, similar to DDT and its metabolite dichlorodiphenyldichloroethylene (DDE), have long half-lives, they continue to be detected in humans, making their potential toxicity a pressing issue [9].

Anand et al. studied OCP exposure in placental samples from 40 women who delivered at 37 weeks or less and from 50 women who delivered at a date beyond 37 weeks [10]. This population of Indian women was considered at-risk because they lived in a populous state in which OCP use has remained high, despite national bans against agricultural use [10]. They found that concentrations of DDT, DDE, and hexachlorocyclohexane isomers were significantly higher in PTBs compared to those in controls [10].

An earlier study from the US Collaborative Perinatal Project (CPP) examined 2380 children born between 1959 and 1966 and measured the DDE levels in maternal serum samples stored during pregnancy [11]. The concentration of maternal DDE was significantly correlated with the odds ratio of PTB, and a direct relationship between concentration and PTB incidence was established. The CPP authors concluded that their findings strongly suggest that DDT use increases PTB, and that this information should be a factor in the decision-making process for future use of DDT [11].

In contrast to Anand et al. and the CPP study, an independent study found no significant association between OCP exposure and PTB [9]. They studied 420 births in a cohort of women recruited from 1959–1967, a period in which DDT use was rampant. Of the 420 cases, 33 were born prematurely, and an inverse, albeit nonsignificant, relationship between OCP exposure and PTB was observed. The conflicting findings, compared with studies that reported a significant link between PTB and OCP levels, were attributed to population differences. They assessed that women seeking health care from teaching hospitals may be different from those seeking care through health insurance because the former may represent lower-income populations that experience different environmental exposures [9].

Collectively, these studies provide strong evidence for an association between OCP exposure and PTB and highlight the need for future studies that include larger sample sizes. Because both studies included relatively small sample sizes, future studies should aim to recruit a greater number of participants. Additionally, the two populations discussed were vastly different, with one living in a country in which DDT use is illegal and the other living in an area with high levels of OCP exposure with continued DDT use [9,10]. While the different results could be due to a variety of factors, the level of exposure needed to increase PTB risk could be a future topic of investigation.

### 5.2. Organophosphate Pesticides (OPPs)

In addition to OCPs, OPPs are widely used. As the toxic effects of DDT became clear, a well-known OPP, chlorpyrifos, was invented as an alternative. Despite several countries banning DDT, chlorpyrifos and other OPPs are still in use [26]. Today, OPPs are used in commercial agriculture and home gardens to control pests. OPPs inhibit cholinesterase, an enzyme that breaks down the neurotransmitter acetylcholine [12]. Thus, these agents disrupt neuroendocrine signaling and have neurotoxic effects.

Eskenazi et al. assessed OPP metabolites in urine samples from pregnant women living in an agricultural population, in which approximately 500,000 pounds of OPPs are applied annually [12]. Levels of dialkyl phosphate metabolites in maternal urine during pregnancy, pesticide-specific metabolites during pregnancy, and cholinesterase were measured. The authors found that gestational length was negatively associated with metabolites of diethyl phosphate and with umbilical cord cholinesterase levels [27]. Because the authors measured urine samples collected at two points during pregnancy, they also found that increasing exposure levels in late pregnancy decreased gestational length to a greater extent than did increasing exposure levels in early pregnancy [27].

Similar to Eskenazi, et al., Jaaks et al. measured OPP exposure in a vulnerable population [14]. Among women in Bangladesh, they found that those with elevated 4-nitrophenol, a metabolite of an OPP, were more than three times more likely to deliver preterm compared to those with low 4-nitrophenol [14]. Both studies acknowledged that they measured pesticides and metabolites individually, so they were not aware of potential interactions [14,27]. Another study by Wang et al. found a significant inverse association between diethyl phosphate metabolites and gestational length among females, but not males [15].

Collectively, these studies suggest that OPPs can increase the risk of PTB, although the effect may depend on the timing of exposure and gender of the fetus [14,15,27]. Future analyses can attempt to clarify this finding while also specifying whether specific OPPs increase risk of PTB. Eskenazi et al. and Wang et al. both measured dialkyl phosphate metabolites to assess exposure to OPPs, in general [15,27]. In contrast, Jaaks et al. measured levels of eight specific biomarkers [14]. Conducting similar analyses would inform whether some OPPs are safer than others are, in regard to their exposure and associated PTB risk.

## 6. Chemicals in Products

Chemicals in products, such as lead and flame retardants, are EDCs that are present in common household items, including electronics, personal care products, children’s products, and clothing. These products are dangerous on two fronts, both of which underscore the importance of understanding their harmful consequences. First, lead and flame retardants are essentially necessities in today’s world, so avoiding the exposure to these agents and their EDC properties may not be practical. Second, corporations often fail to disclose the chemical composition of their products, making consumers unaware of what they expose themselves to [1].

### 6.1. Lead

Lead exposure continues to be a public health concern around the world, as it can be a predictor of adverse health outcomes, such as cognitive decline, cardiovascular disease, and decreased fetal growth [4]. While exogenous lead exposure through polluted air, food, drinking water, and cigarette smoking itself causes damage, endogenous exposure can also lead to harmful consequences [4,16,17]. Endogenous lead can remain in bones for decades, and it can easily cross the placenta in pregnant women [17,24].

Cantonwine et al. attempted not only to corroborate previous findings associating lead exposure and PTB, but they also studied whether certain stages of pregnancy were more vulnerable to the effects of lead on PTB [4]. Their results showed that plasma lead levels in the first and second trimesters were significantly associated with decreased gestational age, but not with PTB. The authors attributed the lack of association between PTB and lead levels to the low number of PTBs (22) among the study population and the estimation of gestational age by recall of last menstrual period [4].

In addition to investigating the relationship between gestational age and effects of lead, Li et al. classified their participants into groups based on their plasma lead levels (low, medium, and high) to further specify the relationship between lead exposure and PTB [16]. Similar to many previous studies, they found that maternal serum lead level was positively associated with PTB risk. Additionally, they found that the adjusted odds ratio for PTB was higher among subjects with medium lead exposure levels in the first trimester than among those in the second trimester [16]. The same relationship was found among participants with high lead exposure. Both results suggest that PTB risk increases with earlier lead exposure. A strength of this study was the large number of PTB cases (177), while the limitations were the inability to examine lead exposure separate from other potential exposures and the fact that serum lead levels at different gestational ages were not measured among the same women [16].

In contrast to the previously discussed studies, Perkins et al. had different findings after examining the effects of low lead exposure levels on PTB [17]. The mean lead level among women in this sample was 0.4 μg/dL, compared to the average of US females of 0.6 μg/dL, a level at which the effects of lead are not well-studied. In their overall analysis, they did not find a significant association between lead levels and PTB risk [17]. However, after stratifying based on sex, they found that gestational age was inversely associated with lead levels among males. The authors believe their findings have low generalizability, but the differing findings between gender provide reason for further research [17].

In summary, while it has been difficult to establish a significant correlation between lead exposure and PTB, several studies have described a negative relationship between lead exposure and gestational age. Although Cantonwine et al. believed their low-PTB sample size contributed to the lack of significant association between PTB and lead exposure, their results still suggested that lead exposure can decrease gestational age [4]. Li et al. supported their belief, as they reported a significant association between PTB and lead exposure among their large pool of PTB cases [16]. While these authors studied the effects of elevated lead levels, Perkins et al. were one of the first groups to research whether common lead levels increased PTB risk [17]. Though their overall findings did not show significant results, their analyses following stratification based on sex suggested that even low levels of lead can decrease gestational age among males. Collectively, because evidence exists that both high and low levels of lead may increase risk of PTB, future studies may attempt to build on the work of Perkins et al., as the levels of lead exposure they studied are more common among women.

### 6.2. Polybrominated Diphenyl Ethers (PBDEs)

PBDEs have been widely used as flame retardants in home construction, furniture, clothing, and electronic appliances for decades. Similar to other EDCs, they can leach into the environment and enter bodies through inhalation or ingestion. Just as lead remains in bones for years, PBDEs accumulate in lipophilic tissues, creating a route of endogenous exposure [28].

Peltier et al. studied the effect of PBDE-47, the most common form of PBDE, exposure on PTB risk [28]. After analyzing serum samples that were collected at the time of delivery, they found that PBDE-47 concentrations were higher for a sample of American women who delivered prematurely. In contrast, Gao et al. found that another form of PBDE, PBDE-153, was significantly associated with PTB among Chinese women living in an area that produces a large amount of flame retardant [29].

Chen et al. also studied a sample of Chinese women from this same high-production area, but they did not find a relationship between PBDE exposure and PTB [30]. Harley et al. examined a sample of women from an agricultural community in California and, similar to Chen et al., did not find a significant association between PBDEs and PTB [31].

Collectively, these studies show conflicting findings regarding the effect of PBDE exposure on PTB risk, with all acknowledging sample sizes as being limitations [28,29,30,31]. They did not have many PTB cases in their populations, which may have led to a lack of strong associations or the lack of any effect at all. Currently, few studies on the reproductive effects of PBDE in humans exist, and there are even fewer on the relationship between PBDE and PTB [29]. Future studies could consider attempting to measure PBDE exposure longitudinally by collecting samples at various points during the participants’ pregnancies, as opposed to these reviewed studies, which collected single determinations in body fluids [28]. Although PBDEs have long half-lives and are persistent in individuals, pregnancy physiology and diet pattern may alter PBDE levels during pregnancy. As a result, an association between EDC levels during different trimesters and fetal developmental stages may reveal information regarding windows of gestational sensitivity.

## 7. Discussion

Though there are varying findings, a significant number of studies suggest an association between EDC exposure and increasing PTB risk. Because of the inability to conduct randomized controlled trials to study EDCs, it is difficult to draw a direct cause-and-effect relationship between EDC exposure and PTB. With the ubiquitous nature of EDCs and multiple routes of exposure, there are no ethical ways to expose participants to a single EDC to study its effect in isolation (Table 1). Therefore, participants, who have likely been exposed to a variety of EDCs, must be randomly recruited. As a result, regulatory policies restricting or prohibiting the use of EDCs can be difficult to enact. However, governments, especially those trying to lower high PTB rates, should consider the large number of studies reporting inverse relationships between EDCs and gestational age as evidence for regulations against EDCs.

Among the ten countries with the highest rates of PTB per 100 live births, eight are in Africa (Table 2). Furthermore, the highest six rates belong to African countries [3]. With the rapid urbanization and economic development that many African countries are currently undergoing, the exposure to EDCs stemming from products such as cell phones, laptop computers, and personal care products is increasing [32]. Despite the quickly transforming economy, agriculture remains the largest economic sector in many of these countries and the largest user of EDCs such as pesticides [32]. The evolving economy coupled with the traditional staple of agriculture make for a potentially harmful environment. Hence, African countries, along with many others, have focused on improving EDC policies. African delegates have participated in meetings for the Strategic Approach to International Chemicals Management (SAICM), a policy framework administered by the UNEP that aims to bring countries together on the issues of EDCs [32,33]. The goal of the SAICM is to create a future in which chemicals will be produced and used in ways that limit harm done to humans and the environment [33].

The WHO reported on the engagement of the health sector in SAICM, as of 2015 [34]. WHO and UNEP have led global efforts to raise awareness and develop guidelines about the use of lead in paint, EDCs, nanotechnology and manufactured nanomaterials, and hazardous substances within the life cycle of electronic products, otherwise known as “e-waste.” In addition to global efforts, WHO regional offices, such as those in Africa, Europe, Latin America, Asia-Pacific, and the Eastern Mediterranean, study the more specific patterns of EDC use and areas for improvement in each locale [34]. For example, the WHO offices in Africa conducted an assessment that found that there is a lack of a mechanism for coordination among the various sectors that are responsible for chemical management [34]. European offices outlined six areas that need to be focused on for the health sector to make meaningful change in EDC use: awareness raising, risk assessment, capacity building, information collections and dissemination, intersectoral communication, and international leadership. Similarly, the Asia-Pacific region concluded that multi-sectorial approaches, improved collaboration with academic institutions, and improved networking with poison centers in the region are necessary [34].

Many factors, such as nutrition, access to clean water, medical care, and infection control, underlie the high PTB prevalence in countries throughout the world. The improvement of EDC policies alone will likely not dramatically reduce PTB rates. However, in a country such as Malawi, where there are about 18 PTBs per 100 live births, the urgent nature of this issue emphasizes the need for rapid changes, one of which could be regulation of the production and use of EDCs, in addition to addressing problems that may be more entrenched [3] (Table 2). In the future, the work of the WHO, UNEP, and countries around the world should continue their work to better understand the effects of EDCs and how they can create environments in which humans and their surroundings are protected from the harmful effects of EDCs.

## Figures and Tables

**Table 1 ijms-24-01952-t001:** Routes of EDC exposure and ability to reach fetus.

Source	Chemical	Used In	Routes of Exposure	Is Fetus Exposed?
Food Contact Materials	BPA	Reusable food and drink containers, food can linings, water bottles, dental sealants, thermal receipts, medical equipment, flooring, water supply pipes [5]	Topical, oral, inhalation [5,6]	Yes [5]
	Phthalates	Reusable food and drink containers, lotions, perfumes, deodorants [7]	Topical, oral [7]	Yes [8]
Pesticides	OCP	Pesticides [8]	Topical, oral, inhalation [8,9]	Yes [10]
	OPP	Pesticides [11]	Topical, oral, inhalation [8,9]	Yes [12]
Chemicals in Products	Lead	Paint, children’s products, drinking water, cigarettes [13,14,15]	Oral, inhalation, endogenous (bones) [4,14,15]	Yes [4]
	PBDE	Furniture, clothing, electronic appliances [16]	Oral, inhalation, endogenous (adipose) [16]	Yes [17]

**Table 2 ijms-24-01952-t002:** Countries with highest rates of PTB per 100 live births [3].

Country	PTB Rate per 100 Live Births
Malawi	18.1
Comoros	16.7
Congo	16.7
Zimbabwe	16.6
Equatorial Guinea	16.5
Mozambique	16.4
Gabon	16.3
Pakistan	15.8
Indonesia	15.5
Mauritania	15.4

## Data Availability

No new data were created or analyzed in this study. Data sharing is not applicable to this article.

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
