# Peer review of "Association of Preterm Birth and Exposure to Endocrine Disrupting Chemicals"

_ijms, 2023, doi:10.3390/ijms24031952_

Round 1

Reviewer 1 Report

The manuscript describes the evidence on EDC exposure and preterm birth, with a pitch to global health. The search strategies are not clear, and the inclusion of relevant literature is not exhaustive. It may function more as a scoping review rather than a comprehensive review. Comments are provided below for the authors to consider in the revisions.

11.       Line 146-148: the sentence showing higher urinary BPA concentrations in mothers delivered PTB was duplicating the meaning of Line 142=144. One of them can be deleted.

22.       Line 157: medically necessary PTBs can be termed as indicated PTBs.

33.       Line 159-161: Indicated PTBs were delivered following obstetric guidelines, but it is not always true to state “not because of pathology. The authors believed these births likely would have been carried to term without intervention.” The authors need to differentiate preterm birth subtypes that need to be delivered to reduce maternal and fetal complications. If not delivering, the condition is detrimental to maternal or child health. There are also PTB on maternal request, in which mothers want to deliver earlier without medical indications. Most countries tend to prohibit delivery without medical indications.

44.       Line 166-167: the sentence can be deleted.

55.       Line 172 to 177: it is confusing whether the Huang study found an association. Line 172 stated “positively associated with PTB,” while Line 176 stated “inconsistent findings between their study and those that….”

66.       Line 233: The following paper has been published using 16 cohorts to examine phthalates and PTB. https://pubmed.ncbi.nlm.nih.gov/35816333/

77.       Line 272: This statement needs to be rephrased to indicate inverse but non-significant association.

88.       Line 279: In the literature, an earlier study on DDE and PTB can be mentioned. https://pubmed.ncbi.nlm.nih.gov/11463412/

99.       Line 321: It is not true that “chemicals in products, such as flame retardants” are “essentially necessities in today’s world.” If the authors mean the products are essential, that can be arguable. There is no reason to state the chemicals are essential in the products.

110.   Line 337: if the intention was to show low number of PTB cases, only 22 is needed. The 9.4% is not low in proportion.

111.   Line 348: “the lack of exclusion of other metals” needs to be rephrased to indicate other metals are not measured or not associated with lead or outcome.

212.   Line 359-361: This sentence can be rephrased to indicate inverse association between lead exposure and gestational age.

113.   Line 394-397: it is not true that PBDEs are measured in “single urine samples” as majority of studies measured PBDEs in serum or plasma samples. The need to measure PBDE exposure longitudinally is low as PBDEs are persistent and have long half-lives.

114.   Line 441: It is probably necessary to point out that to reduce PTB rates in developing countries, multi-prong approaches are needed to improve nutrition, control infections, reduce environmental exposures (EDC is among them, others are indoor air pollution, outdoor air pollution, drinking water contaminations, etc.)

Author Response

Reviewer 1

We thank Reviewer 1 for their time and excellent suggestions for improving the quality of the manuscript. We especially appreciate the recommendations for clarity in rewording and in offering some additional valuable references. The comments have been addressed in the revised manuscript as described below:

Reviewer 1: The manuscript describes the evidence on EDC exposure and preterm birth, with a pitch to global health. The search strategies are not clear, and the inclusion of relevant literature is not exhaustive. It may function more as a scoping review rather than a comprehensive review. Comments are provided below for the authors to consider in the revisions.

Author Response: The search strategies utilized are now included in the abstract.

Reviewer 1: Line 146-148: the sentence showing higher urinary BPA concentrations in mothers delivered PTB was duplicating the meaning of Line 142=144. One of them can be deleted.

Author Response: The 1st reference to the higher urinary BPA concentration in mothers with PTBs has been amended to introduce the study, and the conclusion has been deleted as suggested.

Reviewer 1: Line 157: medically necessary PTBs can be termed as indicated PTBs.

Author Response: The phrase ‘medically necessary PTBs’ was corrected to read ‘indicated PTBs.’

Reviewer 1: Line 159-161: Indicated PTBs were delivered following obstetric guidelines, but it is not always true to state “not because of pathology. The authors believed these births likely would have been carried to term without intervention.” The authors need to differentiate preterm birth subtypes that need to be delivered to reduce maternal and fetal complications. If not delivering, the condition is detrimental to maternal or child health. There are also PTB on maternal request, in which mothers want to deliver earlier without medical indications. Most countries tend to prohibit delivery without medical indications.

Author Response: This section was revised to now more clearly state that the referred preterm birth subtypes where those that needed to be delivered to reduce maternal and fetal complications.

Reviewer 1: Line 166-167: the sentence can be deleted.

Author Response: The sentence “Because of the findings in their secondary analysis and study limitations, the authors believed the association between BPA exposure and PTB warrants further research” was deleted.

Reviewer 1: Line 172 to 177: it is confusing whether the Huang study found an association. Line 172 stated “positively associated with PTB,” while Line 176 stated “inconsistent findings between their study and those that….”

Author Response: This section was revised to clarify that an association was indeed found, and Line 176 was revised to state “They believed the sample size, study design, and differences in biomarkers contributed to the low incidence of PTB reported.”

Reviewer 1: Line 233: The following paper has been published using 16 cohorts to examine phthalates and PTB. https://pubmed.ncbi.nlm.nih.gov/35816333/

Author Response: A new paragraph (see below) and reference has been added to the manuscript, which discusses the results reported in the reference suggested by Reviewer 1.

Recently, the association between phthalate exposure and PTB was examined using 16 cohorts of individuals living in the united states. Urinary phthalate metabolite analysis was used as a marker of phthalate exposure. The studies revealed that the odds of PTB was 12% to 16% higher in individuals with increased urinary concentrations of MEBP, mono-isobutyl phthalate, mono(2-ethyl-5-carboxypentyl) phthalate, and mono(3-carboxypropyl) phthalate. Furthermore, the authors estimated that hypothetical reduction of phthalates levels by 10%, 30%, and 50% would have prevented 1.8, 5.9, and 11.1 of the approximately 90 preterm births in the study population, respectively [16].

[16] Walsh, B.M.; Keil, A.P.; Buckley, J.P. et al. Associations Between Prenatal Urinary Biomarkers of Phthalate Exposure and Preterm Birth: A Pooled Study of 16 US Cohorts. JAMA Pediatr. 2022, 176(9), 895-905.

Reviewer 1: Line 272: This statement needs to be rephrased to indicate inverse but non-significant association.

Author Response: The text has been revised to state more clearly that the cited study found an inverse, yet nonsignificant, relationship between OCP and PTB.

Reviewer 1: Line 279: In the literature, an earlier study on DDE and PTB can be mentioned. https://pubmed.ncbi.nlm.nih.gov/11463412/

Author Response: This is a great suggestion, and the following reference and paragraph was added.

An earlier study from the US Collaborative Perinatal Project (CPP) examined 2380 children born between 1959 and 1966 and measured the DDE levels in maternal serum samples stored during pregnancy [20]. The concentration of maternal DDE was significantly correlated with the odds ratio of PTB, and a direct relationship between concentration and PTB incidence was established. The CPP authors concluded that their findings strongly suggest that DDT use increases PTB, and that this information should be a factor in the decision-making process for future use of DDT [20].

[20] Longnecker, M.P.; Klebanoff, M.A.; Zhou, H.; Brock. J.W. Association between maternal serum concentration of the DDT metabolite DDE and preterm and small-for-gestational-age babies at birth. Lancet 2001, 358(9276), 110-114.

Reviewer 1: Line 321: It is not true that “chemicals in products, such as flame retardants” are “essentially necessities in today’s world.” If the authors mean the products are essential, that can be arguable. There is no reason to state the chemicals are essential in the products.

Author Response: The text has been revised to indicate that the products are essential, not the chemicals.

Reviewer 1: Line 337: If the intention was to show low number of PTB cases, only 22 is needed. The 9.4% is not low in proportion.

Author Response: The text has been revised to state ’22,’ and the 9.4% was deleted.

Reviewer 1:   Line 348: “the lack of exclusion of other metals” needs to be rephrased to indicate other metals are not measured or not associated with lead or outcome.

Author Response: The text has been rephrased to read the inability to examine lead exposure separate from potential other exposures”

Reviewer 1: Line 359-361: This sentence can be rephrased to indicate inverse association between lead exposure and gestational age.

Author Response: The text has been rephrased to indicate an inverse association between lead exposure and gestational age.

Reviewer 1: Line 394-397: it is not true that PBDEs are measured in “single urine samples” as majority of studies measured PBDEs in serum or plasma samples. The need to measure PBDE exposure longitudinally is low as PBDEs are persistent and have long half-lives.

Author Response: This section has been revised to state that PBDEs are measured in single determinations of body fluid.

Reviewer 1: Line 441: It is probably necessary to point out that to reduce PTB rates in developing countries, multi-prong approaches are needed to improve nutrition, control infections, reduce environmental exposures (EDC is among them, others are indoor air pollution, outdoor air pollution, drinking water contaminations, etc.)

Author Response: Reviewer 1 provided an outstanding recommendation. We have included “nutrition, access to clean water, medical care, and infection control” as additional factors to consider in lines 517-518.

Reviewer 2 Report

The manuscript entitled "Association of Preterm Birth and Exposure to Endocrine 2 Disrupting Chemicals" presents a very interesting scientific leterature findings regarding the effect of EDC exposure on reproductive systems, (preterm birth). The manuscript is well-written and easy to read. the results are clear and concise and the discussion is also well presented. Therefore, I suggest the article be accepted as submitted after include more informations about how the papers were selected to be use in this review.

Author Response

Author’s Response to Reviewer 2

We thank Reviewer 2 for their exceeding positive review of the manuscript and excellent suggestions for improvement. To address Reviewer 2’s comments, the abstract has been revised to describe how PubMed was used with specific search terms [BPA; phthalates; organochlorine pesticides; organophosphate pesticides; lead; PBDE; preterm birth]. This enabled identification of articles published in the past 20 years that examine the association between EDC exposure and preterm birth.

Round 2

Reviewer 1 Report

Thanks for the revisions. Some minor comments are below for the revised version. 

Line 234: MEBP should be MBP, as the authors used that for mono-n-butyl phthalate.

Line 238: The 90 preterm births need to be in the context of per 1000 live births, otherwise, it is not accurate.

Line 408-413: PBDEs are considered persistent as authors stated. It does not entirely exclude the possibility that pregnancy physiology and diet pattern may change the PBDE levels during pregnancy, however, repeatedly measuring PBDEs during different trimesters may not be needed.
